# A Review of Biochar Properties and Their Utilization in Crop Agriculture and Livestock Production

**Kajetan Kalus** [1,*], **Jacek A. Koziel** [2] **and Sebastian Opaliński** [1]

1    Faculty of Biology and Animal Science, Department of Environment Hygiene and Animal Welfare, Wroclaw
     University of Environmental and Life Sciences, 51-630 Wrocław, Poland
2    Department of Agricultural and Biosystems Engineering, Iowa State University, Ames, IA 50011, USA
*    Correspondence: kajetan.kalus@upwr.edu.pl; Tel.: +48-698-660-458

**Abstract:** When it comes to the use of biochar in agriculture, the majority of research conducted in the last decade has focused on its application as a soil amendment and for soil remediation. This treatment improves soil quality, increases crops yields, and sequestrates atmospheric carbon to the soil. Another widely studied aspect connecting biochar with agriculture is the composting processes of various agricultural waste with the addition of biochar. Obtaining the material via the pyrolysis of agricultural waste, including animal manure, has also been investigated. However, given the remarkable properties of biochar, its application potential could be utilized in other areas not yet thoroughly investigated. This review paper summarizes the last decade of research on biochar and its use in crop agriculture and livestock production. Knowledge gaps are highlighted, such as using biochar for the mitigation of odorous emissions from animal manure and by feeding the biochar to animals.

**Keywords:** biochar; agriculture; livestock production; feed additive; animal nutrition; composting; pyrolysis; torrefaction

## 1. Introduction

Biochar (BC) is obtained via the pyrolysis of organic material. In the process, biomass is treated by heat in very low oxygen (or other oxidizers) content, where complex chemical compounds are transformed into simpler ones, eventually yielding gaseous products such as water vapor, carbon dioxide ($CO_2$), carbon monoxide (CO), hydrogen ($H_2$), methane ($CH_4$), ethane ($C_2H_6$), and also solid carbon residue often named as "char". "Biochar" indicates that the material has been obtained from biomass, such as hay, corn stover, bagasse, switchgrass, rice hulls, woodchips, animal manure, or even sewage sludge [1], with the primary purpose of utilization rather than as an energy source. Many of those materials are byproducts of agricultural production or municipal waste management, often treated or classified as a waste that needs to be disposed. Reusing this waste to produce BC that have outstanding properties and can be utilized in crop agriculture, animal farming, and environmental protection [2] is an excellent way of carbon and energy recovery.

Biochar properties are dependent on the type of feedstock material used to obtain the BC and the conditions under which the pyrolysis is conducted, especially its temperature and duration of the process. Different types of biomass yield BC of different characterization, but for most of the feedstock materials, a general rule applies—with an increase of pyrolysis temperature, the obtained BC is characterized by higher C content, energy content, porosity and surface area with lower O and H content, and lower BC bulk density [3]. An increase of pyrolysis time increases C content and surface area of BC [4].

When it comes to the use of BC in agriculture, a subject that was thoroughly investigated by research during the past decade is the use of BC as a soil amendment. Parameters that are dependent on each other, such as the improvement of soil quality and microbial activity, the increase in crop yields and the reduction of greenhouse gases emissions associated with the extensive use of fertilizers, have been intensively investigated. In this paper, the last decade of research on BC in agriculture is summarized and also new possible ways of utilizing BC, that are recently emerging as research subjects, especially in the field of livestock production, are reviewed.

A summary of the scope and content of this review paper is presented in Figure 1. Information is summarized in tables at the end of each section.

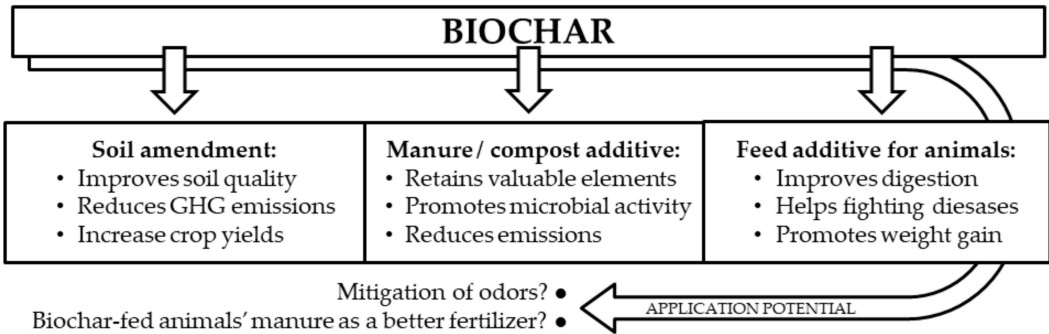

**Figure 1.** Summary of the scope and content of review focused on the use of biochar in agriculture.

## 2. Biochar Properties

Properties of biochar have been very well described by Weber and Quicker [3] in their review article. In this section, the most important properties of BC from the perspective of agriculture and livestock production have been discussed. BC characteristics are determined by its composition, which depends on the feedstock material used for the pyrolysis and the process parameters themselves, with the feedstock material being the most significant factor [5]. In the process of BC production, the main interest is to obtain the most of a carbonaceous solid-state product. Typical BC pyrolysis time and temperature range from 60 to 240 min and from 300 to 700 °C; however, those parameters are determined by the desired properties of the final material [3]. Torrefaction, a low-temperature pyrolysis at a range of 200 to 300 °C, also referred to as "mild pyrolysis" or "high-temperature drying", has recently gained more interest as it allows a feedstock material with similar properties to be obtained but with simpler technology compared to regular pyrolysis for the same feedstock material, due to the lower temperature of the process [6].

Increasing interest in biochar production has resulted in a demand for a high-quality product and regulatory oversight. The European Biochar Foundation [7], a non-profit institution, provides voluntary quality standards and guidelines to ensure that the manufacturing of BC leads to a material that has satisfactory properties and a positive environmental footprint. The International Biochar Initiative (IBI) [8] is another platform for fostering stakeholder collaboration, best industry practices, and environmental and ethical standards to support biochar systems that are considered safe and economically viable. IBI Biochar Standards provide tools to confirm that biochar possesses the necessary characteristics and meets quality standards for safe use. This is especially important for agricultural uses of BC that could be potentially contaminated with toxics (e.g., heavy metals, polycyclic aromatic hydrocarbons, dioxins, volatile and semivolatile organic compounds). Biochars of unknown origin or produced from contaminated waste substrates are more likely to be of concern, compared to the substrates used for biochar production in agriculture, such as corn stover, pruning residues, or poultry litter.

Mahdi et al. [4] investigated the influence of pyrolysis conditions on date seed BC properties. An increase in pyrolysis temperature and heating time of the same feedstock material, from 350 to 550 °C and from 1 to 3 h, resulted in a BC characterized by higher C content (from 64.4% to 82.2%), ash

content (from 6.67% to 12.62%) and pH (from 6.95 to 8.58); however, at the same time, BC yield was reduced, from 45.18% to 20.94%.

Kim et al. [9] used pitch pine woodchips during a fast pyrolysis (2 s) under different temperatures (300, 400, and 500 °C) and the increased temperature resulted in a BC with higher C content (63.9%, 70.7%, and 90.5% respectively), but lower H (5.4%, 3.4% and 2.5% respectively) and O (30.4%, 25.5%, and 6.7% respectively) content. BC yields were also decreased with the increase of pyrolysis temperature, from 60.7% at 300 °C, 33.5% at 400 °C, to 14.4% at 500 °C.

Arni [10] compared effects of slow (heating rate of 50 °C/min, 60 min of the process) and fast (heating rate of 120 °C/min, 20 min of the process) pyrolysis of sugarcane bagasse and proved that slow pyrolysis yields a fuel of a higher char content than the fuel obtained from the same material with fast pyrolysis. Fuel obtained in the process of fast pyrolysis was characterized by higher fluid fraction (tar) content from 50.89% at 480 °C to 38.11% at 780 °C compared to the slow pyrolysis, where tar content ranged from 27.11% at 390 °C to 19.20% at 980 °C. Char content was respectively higher (slow pyrolysis) and lower (fast pyrolysis), but no linear relationship between char content and pyrolysis temperature can be stated. The higher char content for the slow pyrolysis is likely an effect of secondary char formation, which occurs in slower processes. However, the general trend can be observed that no matter the type of pyrolysis, the increase of its temperature leads to lower char content, which means a lower yield of BC.

Research by Hossain et al. [11] showed that an increase in temperature of wastewater sludge pyrolysis results in lower BC yields from 72.3% for pyrolysis at 300 °C down to 52.4% at 700 °C. However, the increase in pyrolysis temperature resulted, on the contrary, in a decrease of the C content from 25.6 to 20.2% between 300 and 400 °C, and then remained stable. Such a change in C content could be explained by the high content of C-containing volatile matter of the feedstock material that was easily gasified with the increase in temperature. H, N, and O contents also decreased with the higher temperature, from 2.55% to 0.51%, 3.32% to 1.20%, and 8.33% to ~0.00%, respectively.

Research by Song and Guo [12] also showed that an increase in the temperature of poultry litter pyrolysis from 300 to 600 °C leads to obtaining BC with lower C content (from 38% to 32.5%) and higher ash content (from 48% to 60%), and also lowers the yields from 60% to 45%. In this case, the lower C content of BC is explained by the high ash content of the feedstock material itself.

Besides the properties of the pyrolysis process, different types of biomass used to obtain BC also contribute to its different characteristics. Spokas and Reicosky [13] compared the physical properties of 16 different BC-based commercial products obtained from different source materials at different temperatures ranging from 450 to 815 °C. For feedstock materials pyrolyzed in the temperature range of 450 to 465 °C, the BCs C content ranged from 42.7% for "Biosource$^{TM}$" to 83.0% for coconut shell BCs. Moisture content ranged from 5.3% for pine wood chip to 15.0% for "Biosource$^{TM}$" BCs. Biochars obtained from pyrolysis at higher temperatures ranging from 481 to 515 °C were characterized by C content from 24.6% to 65.7% for both corn stover BCs obtained from different sources. Moisture content ranged from 2.7% for both corn stover BCs, to 16.2% for peanut hull BC. The lower C content of the BC is almost always associated with higher ash content. However, in this case of comparison between BCs from different feedstock materials, higher pyrolysis temperatures do not always result in a BC with higher C content, as the composition of feedstock material itself plays a very important role, and the properties of BCs obtained at similar temperatures but from different materials can differ greatly, e.g., in C content.

Oh et al. [14] investigated the chemical properties of orange peels, residual wood, and water-treatment sludge used as a feedstock material for obtaining BC. The contents of the most important elements were very differentiated. C content in those materials was, respectively, 41.85%, 45.78%, and 6.58%; N content 0.99%, 0.28%, and 1.78%; O content 47.93%, 44.06%, and 15.28%; H content 6.37%, 6.11%, and 1.78%. Different feedstock materials resulted in different properties of BC-based fertilizers. Orange peels, residual wood, and water-treatment sludge BCs obtained through pyrolysis at 300 °C were characterized by lower pH (8.00, 7.80, and 5.80, respectively), higher N content

(1.78%, 1.21%, and 0.23%, respectively) and lower ash content (3.90%, 7.84%, and 82.04%, respectively) compared to the same feedstock material BCs pyrolyzed at 700 °C—pH (12.30, 10.30, and 6.80), N content (0.67%, 0.45%, and 0.13%) and ash content (8.45%, 12.57%, and 90.70%), respectively.

Białowiec et al. [15] obtained BC through the torrefaction of municipal waste and sawdust at temperatures from 200 to 300 °C at 1 h and investigated the effect of temperature on the fuel properties of the final products. An increase of the torrefaction temperature contributed to the decrease of the final product mass yield, volatile solids content, and H/C ratio, while moisture, ash, and sulfur contents increased. Biochar mass yield decreased with temperature for both of the feedstock materials, from 89% at 200 °C to 63% at 260 °C for municipal waste and from 94% at 200 °C to 30% at 300 °C for sawdust; VOCs content decreased from 75% to 57% and from 77% to 40%, respectively and the H/C ratio dropped from 1.4 to 1.0 and from 1.3 to 0.4, respectively. The moisture content of waste BC ranged from 1.4% to 1.9% with no correlation with temperature, while sawdust BC moisture content ranged from 1.5% to 6.0% for 200 and 260 °C, respectively. Any unusual increase in moisture content with a temperature increase might be caused by the high moisture of the feedstock materials and its condensation in the reactor while the material is in its cooling phase. Ash content increased from 14% at 200 °C to 24% at 260 °C for waste BC and from 0.65% at 200 °C to 1.7% at 300 °C for sawdust BC, while sulfur content increased from 0.2% to 0.5% and from ~0.02% to 0.05%, respectively. In case of the waste-derived BC, a temperature of 260 °C (instead of 300 °C) shows the most extreme values, probably due to the high standard deviations, that should be attributed to the high heterogeneity of the feedstock material. The fact that values slightly increase when temperatures exceed 260 °C also have to be noted. In another research, Białowiec et al. [16] investigated properties of municipal waste BC, obtained at 260 °C and 50 min of the torrefaction process. The BC was characterized by C, H, N, O contents of 59.7%, 6.07%, 0.68% and 13.24%, respectively. Moisture and ash contents were 1.54% and 20.14%, respectively.

Świechowski et al. [17,18] examined properties of BC obtained from torrefaction of Oxytree (*Paulownia Clon* in Vitro *112*) pruning residues as a function of torrefaction temperature and time, with the process temperature also ranging from 200 to 300 °C and time ranging from 20 to 60 min. Biochar yields ranged from 52% to 98%, with the lowest yield at the highest temperature and the longest time of torrefaction. H, N, and O contents followed the same trend with the percentage ranging from 3.20% to 7.17%, 1.89% to 4.56%, and 14.90% to 42.20%, respectively. Moisture and C contents followed the opposite trend, ranging from 1.22% to 7.58% and 38.1% to 57.0%, respectively. However, there are some exceptions to the general trends (e.g., lower yields at higher temperatures and higher C content at the same time) that concern different types of cultivation of the feedstock trees.

Pulka et al. [19] torrefied sewage sludge for 60 min at different temperatures ranging from 200 to 300 °C and obtained BC that was characterized by C, H, N, O contents ranging from 28.4% to 12.7%, 2.43% to 0.71%, 4.03% to 2.68%, and 21.31% to 9.02%, respectively, for temperatures from 200 to 300 °C. Ash content increased with temperature from 43.6% to 73.4%. Moisture content was not affected by temperature, but it was reduced when compared to the raw feedstock material.

Syguła et al. [20] used mushroom spent compost as a feedstock material subject to torrefaction at a similar temperature and time range (200 to 300 °C, 20–60 min). The research was focused mainly on the evaluation of torrefaction kinetics, but it is another example of "waste-to-carbon" strategy, showing that wastes of different origin can be successfully turned into a valuable resource.

An extensive analysis of different BC characteristics can also be found in the work of Xie et al. [21]. The general rule applies that the relative content of elements in BC is proportional to the content in the feedstock material, and that an increase in pyrolysis time and temperature leads to BC with higher C content but with lower yields (Table 1). However, there are exceptions to the rule (e.g., for pyrolysis of sewage sludge BC, where the increase in temperature results in lower C content) and thus a complex evaluation of BC properties is often needed when working with a new type of material. The ability to simply change the pyrolysis/torrefaction parameters or using different feedstock material gives the possibility of obtaining carbonaceous material with different, desired properties. The diversity of the

BC properties and the relative ease of their modification makes it an excellent material with application potential that could be utilized in different areas [22]. These include soil amendment, enhancement of the composting process, animal nutrition, odor, gaseous emissions mitigation, and the use of BC-fed animal manure as fertilizers, and many others, with the last three fields not yet thoroughly investigated.

**Table 1.** Summary of selected data on biochar properties.

| Reference | Feedstock Material | Pyrolysis Temperature | C | H | O | N | pH | Ash | Yield |
|---|---|---|---|---|---|---|---|---|---|
| Mahdi et al. [4] | Date seeds | 350 | 64.4 | - | - | - | 6.9 | 6.7 | 45 |
| | | 550 | 82.2 | - | - | - | 8.6 | - | 21 |
| Kim et al. [9] | Pine woodchips | 300 | 63.9 | 5.4 | 30.4 | - | - | - | 61 |
| | | 500 | 90.5 | 2.5 | 6.7 | - | - | - | 14 |
| Świechowski et al. [18] | Oxytree prunings | 200 | 38.1 | 7.2 | 42.4 | 4.6 | - | - | 98 |
| Spokas and Reicosky [13] | BiosourceTM | 450–465 | 42.7 | - | - | - | - | - | - |
| | Pine woodchips | 450–465 | - | - | - | - | - | - | - |
| | Coconut shells | 450–465 | 83.0 | - | - | - | - | - | - |
| | Corn stover | 481–515 | 24.6 65.7 | - | - | - | - | 2.7 | - |
| | Peanut hulls | 481–515 | - | - | - | - | - | 16 | - |
| Oh et al. [14] | Orange peels | 300 | 41.9 * | 6.4 * | 47.9 * | 1.8 | 8.0 | 3.9 | - |
| | | 700 | - | - | - | 0.7 | 12.3 | - | - |
| | Residual wood | 300 | 45.8 * | 6.1 * | 44.1 * | 1.2 | 7.8 | 7.8 | - |
| | | 700 | - | - | - | 0.5 | 10.3 | - | - |
| | Water-treatment sludge | 300 | 6.6 * | 1.8 * | 15.3 * | 0.2 | 5.8 | 82 | - |
| | | 700 | | | | | | | |
| Białowiec et al. [15] | Sawdust | 200 | - | - | - | - | - | 0.7 | 94 |
| | | 300 | - | - | - | - | - | 1.7 | 30 |
| | Municipal waste | 200 | - | - | - | - | - | 14 | 89 |
| | | 300 | - | - | - | - | - | 23 | 63 |
| Białowiec et al. [16] | Municipal waste | 260 | 59.7 | 6.1 | 13.2 | 0.7 | - | 20 | - |
| Hossain et al. [9] | Wastewater sludge ** | 300 | 25.6 | 2.6 | 8.3 | 3.3 | - | - | 72 |
| | | 700 | 20.2 | 0.5 | 0.0 | 1.2 | - | - | 52 |
| Pulka et al. [19] | Sewage sludge ** | 200 | 28.4 | 2.4 | 21.3 | 4.0 | - | 43 | - |
| | | 300 | 12.7 | 0.7 | 9.0 | 2.7 | - | 73 | - |
| Song and Guo [12] | Poultry litter ** | 300 | 38.0 | - | - | - | - | 48 | 60 |
| Chan et al. [23] | Poultry litter ** | 450 | 38.0 | - | - | 2.0 | 9.9 | - | - |
| | | 550 | 33.0 | - | - | 0.9 | 13.0 | - | - |
| Dudek et al. [24] | Brewer's spent grain | 300 | 55.9 | 5.8 | 24.6 | 4.6 | - | 6.7 | - |

Note: Temperatures listed are in °C. C, H, O, N refers to the content percentage (dry basis) of a particular element in BC. Ash refers to the content percentage (dry basis) of ash in BC. Yield refers to the yield percentage of a pyrolysis process. Values marked with * refer to the content percentage of a particular element in a feedstock material, before pyrolysis. Feedstock materials marked with ** highlights the tendency of the properties that stand out in the general trend observed.

## 3. Biochar as a Soil Amendment

Biochar amendment to the soil has become a very extensively studied subject, and numerous review articles analyzing the matter have been recently published [25–33]. BC has proven to be an excellent way of introducing (as a carrier) and retaining (as an adsorbent) nutrients in the soil, sequestrating atmospheric C to it and improving its microbial activity. Those effects could result in a reduction of greenhouse gases (GHG) emissions associated with the extensive use of fertilizers and in the improvement of the general soil quality that leads to an increase of crop yields (Figure 2). The ability of BC to positively affect those parameters in soil could be potentially transferred to other areas that are not as widely studied.

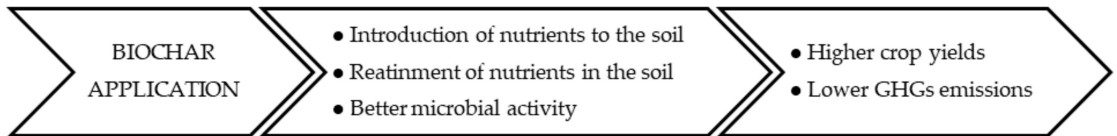

**Figure 2.** Summary of the overall concept of biochar application to the soil.

Chan et al. [23] have stated that there is inadequate knowledge on the properties of BC produced from different feedstock and under different pyrolysis conditions. In their research, they have used the same poultry litter as a feedstock material for two BCs, pyrolyzed in different temperatures (450 and 550 °C) with one BC being activated by high-temperature steam. The BCs were then applied to Alifisol, a typical agricultural soil, at different rates (10, 25, and 50 t·ha$^{-1}$) in order to determine the agronomic values of the BC. In comparison to the control (unamended) group, radish plants grown on the BC-supplemented soils were characterized by a 42% to 95% increase in the total dry matter at a 10 t·ha$^{-1}$ to 50 t·ha$^{-1}$ application rate, respectively, and the yield increase was similar between both BCs. Addition of BCs also significantly changed the plant elemental composition, with increased N, P, S, Na, Ca, and Mg concentrations. Application of poultry litter BCs also changed all chemical parameters of the soils—increasing electrical conductivity, pH, total N and C, Colwell P, exchangeable cations (Ca, Mg, Na, and K), and effective cation exchange capacity but decreasing exchangeable Al. The influence of both BCs on plant yields and their elemental composition is due to the ability of BCs to increase nutrient availability. Comparing the two BCs used in this study, the non-activated BC was more effective than the activated BC. The research also highlights what was stated in the previous section, that different pyrolysis conditions of different feedstock material leads to BCs of different characteristics.

Taghizadeh-Toosi et al. [34] investigated plant bioavailability of BC-adsorbed ammonia by the determination of NH$_3$-BC complex stability. NH$_3$ was isotopically labeled with 15N, and 15N-labelled and unlabeled BC materials were placed into a soil sown with pasture grass seeds, at a BC rate of 1 g per 50 g of the air-dried soil. Four BCs used in the study were manufactured from Monterey pine woodchips at 300, 350, and 500 °C. After 25 d from the addition of the NH$_3$-enriched BCs to the soil, leaf dry matter yields had increased by 2 to 3 times, and root dry matter yields by 2 times, when compared to treatments receiving non-enriched BC. Biochar pyrolyzed in the lowest temperature contributed to the highest N uptake of the plants. However, no differences in plant yield were observed due to the addition of the non-enriched BC. An increase in dry matter yield in the case of the NH$_3$-enriched BC shows that biochar is a very good carrier of a bioavailable N source for plants.

Fellet et al. [35] investigated the amendment of BCs from three different feedstock materials to mine tailings. The effect on the bioavailability of toxic inorganic elements in the tailings and the potential to utilize BC in a phytoremediation process was evaluated. Biochars were obtained from orchard pruning residues pyrolyzed at 500 °C, fir tree pellets, and manure pellets mixed with fir tree pellets; both pyrolyzed at 400 °C. The BCs were mixed with the mine tailings at a rate of 1.5% and 3% (by dry weight), and the mixtures were planted with seedlings of three different plants. After 90 days, the plants were dried, and their shoots and roots were analyzed for the content of toxic metals. In general, the BC application reduced the bioavailability of heavy metals. However, the results are not unequivocal as there is strong significant interaction between the plant species and the dose of BC, regardless of its feedstock material, and while a particular type of BC reduces the bioavailability of one element, at the same time it increases another. The research has shown that BC obtained from manure performs best in the stimulation of plants growth (likely due to the higher N content in the feedstock material) and the immobilization of toxic elements.

Macdonald et al. [36] applied poultry litter and wheat straw BCs at rates of 1, 5, and 10 t·ha$^{-1}$ and investigated the wheat growth response on four different types of soil. The effects of BC amendment were different when considering different types of soil, with a lower total plant biomass in the acidic arenosol at the highest application rate; higher plant biomass in acidic ferralsol, increasing with the

dose of BC applied; and no significant impact at all in neutral vertisol and alkaline calcisol. Thus it is suggested that the use of BC is not limited only to a basic use as a carrier for nutrients and requires a more sophisticated approach concerning soil type in a particular agricultural system that is being amended, with a potential influence on the microbial community of the soil.

Muhammad et al. [37] researched the impact of wheat straw BC on the yield of rice grown on Psammaquent and Plinthudult soils, as well as on the properties of those soils. Biochar was applied to the soils at a rate of 3% (by mass), increasing their pH, total N, and C content, and decreasing the leaching of nutrients from the BC amended soils. The application of BC also increased rice plants growth yield, including their height, leaf length, and grain yield—grain yields were increased by 32% and 41% for Psammaquent and Plinthudult soils, respectively.

The use of BC as a soil amendment is a vast field of study (Table 2) that attracts the attention of a considerable number of researchers. Despite the fact that the subject was primarily examined during the past decade, there is still a lot to investigate, as there are many different relationships between various parameters, such as but not limited to BC feedstock material, dose, and its characteristics, type of soil, plant species, and target elements/compounds of the treatment. Those complex relations can influence differently the overall effect of BC amendment to the soil, and thus still more research is required that include those relationships while investigating each case, in order to achieve a favorable outcome of BC amendment to the soil. However, given the ability of biochar to positively affect those relationships in soil, there is much potential to utilize it in other areas not thoroughly investigated.

**Table 2.** Summary of selected data on biochar application to soil.

| Reference | Feedstock Material | Biochar Dose | Effect |
|---|---|---|---|
| Chan et al. [23] | Poultry litter | 10 to 50 t·ha$^{-1}$ | - Up to 95% increase in plant total dry matter<br>- Increase in N, P, S, Na, Ca, Mg concentrations in plants<br>- Increase in soil electrical conductivity, pH, total N, total C, Colwell P, Ca, Mg, Na, K, and effective cation exchange capacity.<br>- The decrease in soil exchangeable Al. |
| Taghizadeh-Toosi et al. [34] | Pine woodchips | 1 g per 50 g of soil (~20 t·ha$^{-1}$) | - NH$_3$-enriched is a bioavailable N source for plants<br>- Up to 2 times increase in root dry matter<br>- Up to 3 times increase in leaf dry matter |
| Fellet et al. [35] | Pruning residues Fir tree pellets Manure pellets | 1.5% and 3% by dry weight of the soil (~15 and ~30 t·ha$^{-1}$) | - Reduction of heavy metals' bioavailability<br>- Reduction of bioavailability of one element increases the bioavailability of the other |
| Macdonald et al. [36] | Poultry litter Wheat straw | 1.5 and 10 t·ha$^{-1}$ | - Different effects of BC amendment on different types of soil<br>- Lower total plant biomass in acidic arenosol, higher plant biomass in acidic ferralsol |
| Muhammad et al. [37] | Wheat straw | 3% by mass of the soil (~30 t·ha$^{-1}$) | - Increase in pH, total N, total C, and plant growth yield<br>- The decrease in leaching of nutrients |

Note: Values for biochar dose given in parentheses are approximate values in t·ha$^{-1}$, calculated with an assumption that that the biochar has been added to the 10 cm layer of the soil and that the bulk density of the soil is 1 g·cm$^{-3}$.

## 4. Biochar as a Manure/Sludge Additive

One of the environmental aspects of BC utilization is its use as manure, or other compostable waste, additive. Biochar with its high porosity and large surface area can be an excellent adsorbent, retaining valuable elements (e.g., N, C, and S) that typically would be lost to the environment as unwanted pollutants, as well as other microelements in a composting mixture. High surface area

also provides more spacious and aerated habitats for microorganisms, that promote microbial activity additionally stimulated by a relatively high content of less acidic, organic C that is available to the microorganisms [38], thereby having a strong positive influence on composting processes.

Brennan et al. [39] performed a lab-scale experiment to investigate the effect of BC amendment to dairy cattle slurry in order to preserve the slurry nutrients. Wood shavings BC (2 mm dia.) was added to the slurry at a dose equivalent to 3.96 m$^3$ ha$^{-1}$, resulting in 77% NH$_3$, 63% N$_2$O, and 84% CO$_2$ emissions reduction.

Maurer et al. [40] studied the effect of topically applied pinewood BC floating on the surface of swine manure on treating gaseous emissions. Observation showed a 13~23% reduction of NH$_3$ emission with the highest tested BC application dose of 4.56 kg·m$^{-2}$, a 12~30% reduction for H$_2$S at a BC dose of 2.28 kg·m$^{-2}$, and up to a 26% reduction for indole, also with the BC application dose of 2.28 kg·m$-2$. However, the results were not statistically significant. It is worth noting that a 4.56 kg·m$^{-2}$ application dose showed a significant increase of CH$_4$ up to 24%, which is likely due to the addition of labile C and other nutrients for the methanogens present in the manure.

Steiner et al. [41] added 5% and 20% (*w/w*) of pine chips BC to poultry litter that underwent the composting process. Biochar amendment quickened the composting process and emission reductions of H$_2$S and NH$_3$ by 71% and 58%, respectively, were observed for the 20% BC treatment, while 5% BC treatment reduced H$_2$S emission by 58%.

Malińska et al. [42] reported that a 5% amendment of BC obtained from woody material to a sewage sludge–woodchips mixture enhanced organic matter decomposition and reduced NH$_3$ emission by ~50% during the first week of composting.

Janczak et al. [43] achieved ammonia emission reduction of 30% and 44% by adding 5% and 10% (*w/w*) willow woodchips BC, respectively, to a poultry manure–wheat straw mixture.

Research conducted by Wei et al. [44] showed that a 1% BC addition to the composting mixture consisting of poultry manure and tomato stalk had a positive effect on the composting process, with improvements in physicochemical properties and microbial community diversity of the composting mixture.

In the work of Czekała et al. [45], researchers amended a poultry litter–wheat straw composting mixture with 5% and 10% (*w/w*) of >10 mm woody BC. The addition of BC increased the temperature, shortened the thermophilic phase, and increased CO$_2$ emissions by 6.9% and 7.4%, respectively. The authors suggested that the higher CO$_2$ emissions could have resulted from the abiotic oxidation of BC or its available C that functioned as an energy source for microorganisms. The more likely scenario is that the increase in temperature promoted the microbial activity, which resulted in higher CO$_2$ emission from feedstock C digestion. It is very unlikely that C will abiotically oxidize at 70 °C, the higher temperature measured during the experiment.

Agyarko-Mintah et al. [46] investigated green waste BC and poultry litter BC influence on the production of GHGs during composting of poultry litter–straw mixture. The cumulative emission of N$_2$O decreased significantly by 69% and 75% after an addition to the composting mixture of 10% (by weight) of poultry litter BC and green waste BC, respectively. Total N retained at the end of the composting was 40% and 24% higher, respectively. Cumulative CH$_4$ emissions were not significantly different. However, the BC addition contributed to lower CH$_4$ emission on days 15 to 36 of the experiment that may have resulted from the porous nature of BC, which may have improved aeration and methanotrophy. Other authors [47,48] have also confirmed the influence of BC on the composting process and microbial community of composts.

Dudek et al. [24] added brewer's spent grain BC, in the amount ranging from 1% to 50%, to a barley, yeast, hops, and water mixture in order to increase biogas rates, and yield and rate from the anaerobic digestion of the mixture. Maximum production of biogas ranged from 61.0 to 122.0 dm$^3$·kg$^{-1}$$_{dry organic matter}$, with the highest amount for 5% BC addition that resulted in a 32% higher yield, compared to the control group, which coincides with results obtained by Maurer et al. [40].

This is yet another example of a positive way to utilize BC, as the increase in $CH_4$ production during a controlled composting process can lead to the obtainment of a higher amount of green-energy source.

Biochar amendment to the animal manure or other types of wastes subject to composting has a positive effect on the composting process (Table 3), and the positive effects are similar to the ones in the soil, in terms of reduction of greenhouse gases emission or retainment of nutrients. The similarities might apply to other areas of BC utilization. However, comprehensive studies targeting all of the emitted gases should be conducted, if feasible, because, with the reduction of some gaseous emissions, other emissions, especially $CO_2$ and $CH_4$, might increase. Researchers report that the BC addition improves composting process performance and contributes to the mitigation of gaseous emissions overall, retaining precious nutrients in the composted material. However, given the amazing properties of BC, many areas where BC could be utilized remain undiscovered. In the paper of Dias et al. [48], it is merely suggested (in one sentence), that BC could reduce odor emissions from composting processes, and only one paper on the matter has been found. Thus, a complex investigation of BC influence on gaseous emissions is needed, including research on $NH_3$, $H_2S$, and VOCs emissions to evaluate BC amendment deodorizing potential, not only from composting mixtures but, for example, animal manure during livestock production.

**Table 3.** Summary of selected data on biochar application to waste for reduction of gaseous emissions.

| Reference | Feedstock Material | Biochar Dose | Effect |
|---|---|---|---|
| Brennan et al. [39] | Wood shavings | 12% (by volume) of dairy cattle slurry | - 77% $NH_3$ emission reduction<br>- 63% $N_2O$ emission reduction<br>- 84% $CO_2$ emission reduction |
| Maurer et al. [40] | Pinewood | 1.14 to 4.56 kg·m$^{-2}$ of swine manure | - Up to 23% $NH_3$ emission reduction<br>- Up to 30% $H_2S$ emission reduction<br>- Up to 26% indole emission reduction<br>- Up to 24% $CH_4$ emission increase |
| Steiner et al. [41] | Pine chips | 5% and 20% by mass of poultry litter | - 58% $NH_3$ emission reduction<br>- 71% $H_2S$ emission reduction |
| Malińska et al. [42] | Wood | 5% by mass of sewage sludge-woodchips mixture | - 50% $NH_3$ emission reduction during the first week of composting<br>- Enhancement of organic matter decomposition |
| Janczak et al. [43] | Willow woodchips | 5% and 10% by mass of poultry manure-wheat straw mixture | - Up to 44% $NH_3$ emission reduction |
| Wei et al. [44] | Commercially manufactured | 1% by mass of poultry manure-tomato stalk mixture | - Improvements in physicochemical properties of the composting mixture<br>- Promotion of microbial activity in the compost |
| Czekała et al. [45] | Wood | 5% and 10% by mass of poultry manure-wheat straw mixture | - Shortening of thermophilic phase<br>- Increase in temperature of composting mixture<br>- Up to 7.4% increase in $CO_2$ emission |
| Agyarko-Mintah et al. [46] | Green waste Poultry litter | 10% by dry mass of poultry manure-wheat straw mixture | - Up to 75% $N_2O$ emission reduction<br>- Up to 40% of total N retained in the composting mixture |
| Dudek et al. [24] | Brewer's spent grain | 1% to 50% by mass of byproducts of the beer brewing process | - BC can increase biogas production by up to 32%. |

## 5. Biochar as a Feed Additive

The use of char as a feed additive for farm animals have been known for some time, and the results of such supplementation among different groups of animals are, in general, affirmative. This means better digestion, feed conversion ratio, weight gain, or GHGs emission mitigation, and also as a medication against intoxication and bacteriological or viral diseases [49]. Given the above, the use of BC as a dietary inclusion for animal nutrition seems to be a promising solution, and such research direction has recently emerged.

Kultu et al. [50] have shown that the inclusion of oak wood BC into chicken broilers' and laying hens' diets can positively influence their production performance. In the first experiment, 2.5, 5.0, and 10% (*w/w*) BC inclusions to the diet of chicken broilers were examined. Biochar amendment improved the feed conversion ratio, increased body weight gain and also the feed intake of chicken broilers during the first 21 d of the experiment. The 5% BC treatment had the most significant impact with a 23% increase in body weight gain, a 17% increase in feed intake, and a 5% improvement in the feed conversion ratio. Result for days 21 to 42 were not significant. In the second experiment, the inclusion of BC in the starter/finisher diets significantly improved the feed conversion ratio by up to 7%. At the end of the experiment, chicken broilers fed with BC showed significantly higher body weight gain (by up to 8%). In the third experiment, dietary supplementation with BC did not significantly affect any production parameters of laying hens except for the reduction of the number of cracked eggs by 65%, 38%, and 24% for 4%, 2%, and 1% (*w/w*) BC amendment, respectively.

Nageswara Rao et al. [51] investigated the carry-over of aflatoxins into the milk of goats fed with activated charcoal in the amount of 1% of daily dry matter intake. The use of activated charcoal significantly reduced the excretion of aflatoxins by 76% without influencing milk composition, except Zn content.

Research by Van et al. [52] showed that goats supplemented with bamboo BC at the rate of 0.5 and 1.0 g of BC per kg of goat body weight improved dry and organic matter digestibility, while 1.0 g supplementation also reduced N content in urine by 61%. In addition, young goats fed with a BC diet inclusion showed 17% increased daily body mass gain.

Kana et al. [53] investigated growth performance and carcass characteristics of chicken broilers fed with the addition of canarium seed and maize cob BCs (1 mm dia.). For both of the BCs, live body weight and body weight gain were improved with supplementation in a dose of 0.2%, 0.4%, and 0.6% (*w/w*) by up to 17% for 0.2% canarium BC treatment. Charcoal had no significant effect on the intestine circumference, carcass yield, and percentage of the liver, heart, and abdominal fat. Both BC additions increased pancreas weight by up to 120% for a 0.6% dose of canarium BC. Intestine density was not significantly affected by maize cob BC; however, adding 0.2% canarium BC increased intestine density by 20%. An addition of canarium BC in a dose of 0.8% and 1.0% increased creatinine levels by 18 and 86%, respectively. Maize cob BC at a dose of 1.0% also increased the creatinine level by 25%. In addition, maize cob BC depressed the weight of the gizzard by up to 28% for a 0.8% dose while canarium BC did not affect the gizzard size.

Gerlah et al. [54] fed <8 mm diameter charcoal with other supplements to dairy cows at different doses, and investigated the effect of the treatment on blood serum *C. botulinum* ABE and CD antibody levels. A daily application of 400 g of charcoal (1–4 week of the experiment) and 200 g of charcoal with 50 mL of sauerkraut juice (11–14 week of the experiment) per entire herd significantly reduced the antibody levels by about 30% due to the absorption of *C. botulinum* toxins in the gastrointestinal tract by the charcoal.

Evans et al. [55] investigated the effect of poultry litter derived BC as a diet inclusion for chicken broilers. Researchers prepared a mixture of feed and BC (~7% of BC by mass) with a nutrient content similar to one in the regular feed, but where there was no need to include limestone, salt, and tricalcium phosphorus in the mixture. The results of the BC amendment, however, were negative, resulting in 8% higher feed intake, 2% lower weight gain, 11% worse feed conversion ratio, and higher mortality. Such an effect is explained by the high (99 ppm) As content in the BC and its toxic activity. In similar research conducted one year later, Evans et al. [56] investigated 2% and 4% poultry litter BC feed

amendment with lower toxic heavy metals content—birds fed with a 2% BC amendment had increased feed conversion ratio by 7% and birds fed with a 4% amendment had live weight gain decreased by 9%. Those negative effects had to be corrected with phytase inclusion to the feed.

Joseph et al. [57] mixed jarrah wood BC with molasses in a 3:1 ratio and fed it directly to cows. The influence of BC supplementation of diet on cows was not the main subject of the investigation, but no negative effects on animals were reported, and beneficial effects were suggested to have occurred, such as toxin absorption and possibly transformation of organic matter to carbohydrates. Cow dung–BC mixture that was incorporated into the soil had a positive effect on soil properties, which was related to an easier transfer of nutrients from the cow's gut and dung to the soil.

Prasai et al. [58] supplemented layer chicken diet with 4% (by mass) of woody green waste BC and investigated alterations to the overall richness and diversity of the intestinal bacterial community. Supplementation did not affect bird weight, but increased egg productivity by 1.2%, increased average egg weight by 3%, improved feed conversion ratio by 8%, and lowered feed intake by 2%. However, feed amendment with BC did not result in altered diversity and richness of the microbial community within the cloaca microbiota of the laying hens. In another research, Prasai et al. [59] point out that BC is a lesser-known yet promising alternative feed amendment compared to the more popular ones such as bentonite or zeolite. In the research, laying hens were under dietary treatments involving 1%, 2%, and 4% (*w/w*) of woody green waste BC where production parameters and excreta attributes of the hens were investigated. The addition of 2% and 4% of BC to the diet significantly increased egg weight by 5% and 4% respectively. Moreover, a 2% BC amendment significantly increased egg yield by 13%, while feed intake was 7% lower for birds fed twice higher BC concentration and the feed conversion ratio was lower for all the BC treatments by 10%, 14%, and 12%, respectively. For hens of 36 weeks of age, 2% and 4% BC treatments increased egg shell weight by 13% and 4%, respectively, increased shell thickness by 10%, and increased shell breaking strength by 19% but also reduced yolk color score by 19%. For excreta parameters, 2% and 4% BC treatments reduced excreta N content by 17% and 26%, respectively.

Another research by Prasai et al. [60] investigated the effect of woody green waste BC diet inclusion on poultry manure properties. The addition of 1%, 2%, and 4% (by mass) of the BC to the laying hens and chicken broilers diet was made, and their excreta were examined. Increasing contents of BC were associated with a decreased water content of laying hens excreta. Furthermore, 2% and 4% BC amendments significantly lowered manure nitrogen (N) content by 17% and 27%, respectively; however after 35 days of manure decomposition, N losses of the 4% BC treatment were significantly lower and 45% more of N was retained in the excreta. Manure from the 2% and 4% BC amended feed treatments possessed 25% and 45% higher C content, respectively. Ammonia emission was higher from the 2% and 4% BC treatment groups by 47% and 43%, respectively. Manure from BC supplemented broilers, as well as laying hens, was also used to form size class favored granules used for application in fertilizer equipment. Both manures treated with BC produced a significantly higher proportion of granules.

Winders et al. [61] evaluated the effects of a 0.8% and 3% BC diet inclusion for steers on growing and finishing diets. Digestibility, and $CH_4$ and $CO_2$ production from cattle were investigated. Dry matter intake was not affected by BC inclusion in the growing diet but increased quadratically in the finishing diet with the highest increase for the 0.8% biochar amendment. Supplementation with BC improved cattle digestion and also reduced $CH_4$ and $CO_2$ production by up to 9.5% and 9.7%, respectively, for the growing diet and by up to 18.4% and 9.9%, respectively, for the finishing diet. While this research shows that biochar as a feed additive can be used as a mitigator for $CH_4$ and $CO_2$ emissions from cattle breeding, it is interesting that the amendment of composing mixtures with biochar has opposite effect resulting in the production of those gases. A detailed evaluation of the biochar–manure interactions with a distinction to the way of application (directly to the manure or as a feed additive) would be interesting, so as to understand the mechanisms behind such opposite results.

Some similarities between biochar–soil, biochar–compost, and biochar–animal digestive system can be observed—in the case of animal nutrition, BC amendment to the feed also has a potential to

reduce gaseous emission from the manure and provide more nutrients to the animals, resulting in better health and lower feed intakes. Because of the complexity of BC parameters, a detailed evaluation of BC influence on animal health needs to be conducted every time and all the parameters have to be taken into account, including BC particles size, its surface area, C, O, N, H, and also heavy metals (as potential contaminants) content. All the differences in BC properties, as it was in the case of the previously described interactions, can lead to different effects on different animal species, e.g., the same amount of BC diet inclusion was able to increase laying hen production parameters and improve (decrease) the feed conversion ratio, while for chicken broilers the results were the opposite (Table 4). An optimal dose of BC as a feed additive should also be considered because too high a BC inclusion in an animal's diet can lead to digestive problems and thus have a negative influence on the animal's health. However, given that all the BC parameters, including its dose, are optimized, this charred material seems to be a favorable mean of improving the animals' health and their production performance, with more potential fields of utilization. For example, mitigation of odorous volatile organic compound emissions from the manure of livestock with a BC-supplemented diet.

**Table 4.** Summary of selected data on biochar application to animal feed.

| Reference | Feedstock Material | Biochar Dose | Effect |
|---|---|---|---|
| Kultu et al. [50] | Oakwood | 1% to 10% by mass of chicken broilers' and laying hens' feed | - Up to 7% better feed conversion ratio<br>- Up to 23% increase in body weight gain<br>- Up to 65% reduction in the number of cracked eggs |
| Nageswara Rao et al. [51] | Activated charcoal | 1% of goats daily dry matter intake | - 76% reduction in excretion of aflatoxins<br>- Milk composition was not influenced |
| Thanh Van et al. [52] | Bamboo | 0.5 and 1.0 g of BC per kg of goat body weight | - 17% increased daily body mass gain<br>- 61% reduced nitrogen content in urine |
| Kana et al. [53] | Canarium seed Maize cob | Up to 1% by mass of chicken broilers' feed | - Up to 17% increase in body weight gain<br>- Up to 120% increase in pancreas weight<br>- Up to 20% increase in intestine density<br>- Up to 86% increase in creatinine level<br>- Up to 28% reduction in gizzard mass |
| Gerlah et al. [54] | Charcoal | Up to ~1 g of charcoal daily per cow | - Up to 30% reduction in C. botulinum antibody levels |
| Evans et al. [55] | Poultry litter | ~7% of BC by mass of chicken broilers' feed | - 8% higher feed intake<br>- 2% decrease in body weight gain<br>- 11% worse feed conversion ratio |
| Evans et al. [56] | Poultry litter | 2% and 4% by mass of chicken broilers' feed | - 7% better feed conversion ratio<br>- 9% decrease in body weight gain |
| Joseph et al. [57] | Jarrah wood | BC-molasses mixed at a ratio 3:1, fed daily to cows | - Toxin sorption<br>- Easier transfer of nutrients from cow's gut and dung to the soil |
| Prasai et al. [58] | Woody green waste | 4% by mass of laying hens' feed | - 1.2% increase in egg productivity<br>- 3% increase in eggs weight<br>- 8% better feed conversion ratio<br>- 2% lower feed intake |
| Prasai et al. [59] | Woody green waste | 1% to 4% by mass of laying hens' feed | - Up to 5% increase in eggs weight<br>- Up to 13% increase in eggs yield<br>- Up to 7% lower feed intake<br>- Up to 14% better feed conversion ratio<br>- Up to 19% increase in shell breaking strength<br>- Up to 26% reduction of N content in excreta |
| Prasai et al. [60] | Woody green waste | 1% to 4% by mass of laying hens' and chicken broilers' feed | - The decreased water content of laying hens excreta<br>- Up to 27% reduction in excreta N content<br>- Up to 45% higher C content in excreta<br>- Up to 47% higher $NH_3$ emission |
| Winders et al. [61] | Whole pine trees | 0.8% and 3% by mass of feed | - Improved digestion<br>- Up to 18.4% reduction in $CH_4$ emission<br>- Up to 9.7% reduction in $CO_2$ emission |

A broad review article on biochar in animal feeding has been published very recently by Schmidt et al. [62], summarizing the state of knowledge in the matter. Also, after digestion, BC-feed mixtures can become effective manure–BC mixed fertilizers. BC supplementation, compared to other widely used feed additives (e.g., aluminosilicates) is much newer practice, not yet thoroughly investigated but promising. Such a dietary inclusion might also have long-term benefits for both animals and the environment through a chain of consecutive, circulating ways of utilization of (1) biochar as a feed additive, (2) digested biochar-manure mixture as a fertilizer, (3) sequestration of C in the soil as the biochar–manure fertilizer is applied in the fields, and (4) higher crop yields and biomass production to obtain more animal feed and material to produce biochar.

## 6. Conclusions

Until recently, the use of BC in agriculture was mainly focused on the application of BC as a soil amendment. However, there are opportunities to investigate in this wide field of study, as there are plenty potential relationships between various parameters, such as (but not limited to) BC's feedstock material, dose, and its characteristics, type of soil, plant species, and target elements/compounds of the treatment. Another related aspects that were investigated are BC-enhanced composting processes and obtaining the BC via pyrolysis of agricultural waste. However, given the remarkable properties of BC, its application potential could be utilized in other areas such as animal nutrition, odor and gaseous emissions mitigation, and the use of BC-fed animal manure as fertilizers. Biochar amendment to composting has had an overall positive effect on the composting process. However, the reduction of some gaseous emissions might increase others. As an animal feed additive, BC also has a positive influence in general, but the same type of BC can have different effects, even negative, on different animal species. Further studies are required, as this area of BC utilization is relatively new, but promising. Also, different feedstock materials contribute to different properties of BC, and the differences in properties are even more sustained by the conditions of pyrolysis, especially its temperature and time. It is recommended that an optimal dose of BC as an additive should be evaluated, and all influencing BC properties should be taken into the account.

**Author Contributions:** Investigation, K.K.; writing—original draft preparation, K.K.; writing—editing, K.K.; writing—review K.K., J.A.K., S.O.; supervision, J.A.K.; project administration, K.K., S.O.; funding acquisition, K.K., S.O.

**Funding:** (1) Authors would like to thank the Polish National Agency for Academic Exchange for the financial support under "The PROM Programme—International scholarship exchange of Ph.D. candidates and academic staff" co-financed by the European Social Fund. (2) The work is financed by the Wrocław University of Environmental and Life Sciences under the research project number D220/0003/18 (Biochar as a Factor Mitigating the Emission of Odorous Substances from Poultry Production), as part of the "Innovative Doctorate" program. (3) In addition, this project was partially supported by the Iowa Agriculture and Home Economics Experiment Station, Ames, Iowa. Project no. IOW05556 (Future Challenges in Animal Production Systems: Seeking Solutions through Focused Facilitation) sponsored by Hatch Act and State of Iowa funds.

**Conflicts of Interest:** The authors declare no conflict of interest.

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
