# Peer review of "A Review of Biochar Properties and Their Utilization in Crop Agriculture and Livestock Production"

_applsci, doi:10.3390/app9173494_

Round 1
Reviewer 1 Report
The authors present results from some of the latest papers on biochar and its use as soil amendment, as manure and compost additive, and as feed additive for animals.
Dear authors,
Thanks a lot for your interesting draft. In the following you find several specific comments (always with line numbers) and in between some general observations.
0. Abstract
020: These "New areas..." are not that new, but more like described before "not yet thoroughly investigated".
1. Introduction
029: When saying "transformed into simpler components", does this include pyrolysis oil as well?
2. Biochar properties
031: The term "biochar" indicates not only, that it has been obtained from biomass, but also that it is, directly or indirectly, applied to soil in a deliberate manner, with the intent to improve soil properties. This distinguishes it from similar substances like charcoal, which is primarily intended for energy use.
045: Is it the C-"balance" of the soil that was investigated, or rather how the organic carbon content could be increased?
046: These single aims listed here, are not independent from each other. More organic carbon can increase microbial activity, which improves nutrient cycling, which increases yields by reducing nutrient losses to groundwater and air (GHG).
053: A graphic summary of the paper is very thoughtful. However, please reduce the unused white space between the elements and apply a more moderate font-size to "BIOCHAR". In addition, the three bullet items for each chapter, which seem independent from another, are in reality connected. In all three processes, BC supports microbial life, which increases crop yields / weight gain, and reduces emissions (loss), GHG, digestion problems. At least, this should be obvious at the end of the paper. Finally, the last arrow and where it points to (meaning), is open for interpretation.
064: When you write that torrefaction requires a simpler technology than pyrolysis, please be aware that pyrolysis is like the oldest industrial process in history. And that torrefaction requires less energy is only relevant if the feedstock contains not enough of it, which applies only to very wet and low C feedstock.
081: It would be nice to provide an explanation for the lower tar content in slow pyrolysis. If the authors did not mention anything, it might be of secondary char formation, which occurs in slower processes.
086: When you write "also confirms..." it sounds, like it is connected to the previous paper, which it is not (different feedstock).
100: In this paragraph there is a differentiation between lower and higher temperatures. However, it is unclear how many of the 16 source materials were applied to which temperature. Also, you need to specify what 'Biosource' is supposed to be. The reader will not look it up in the original paper.
117: Please specify what kind of "municipal waste" was investigated (also in line 132). If it is not OFMSW, than you should consider if this study belongs in your review (only biochar that can safely be applied to soil or as feed additive).
120: Could you please explain how torrefaction increases the moisture content?
143: Please explain the "exceptions to the trends" or remove this sentence.
151: Please specify "a standard temperature and time range".
152: In regard to torrefaction you speak of "waste-to-carbon" AND "waste-to-energy". However, this is only available for pyrolysis, where you have a solid product (carbon) and a gaseous and sometimes also fluid co-product, which is usable as energy source. Are there such co-products in torrefaction?
162: Table 1: Could you please add a statement regarding dry matter contents? I assume that all data is based on dry matter, but I do not like to assume. Given that all numbers are available here, is it necessary to list them so plentiful in the previous text? Might the readability of the table improve, if you reduce the decimals to a consistent amount? Might it help the reader to arrange similar feedstock materials together, so that similarities and differences are easier to spot?
SUMMARY of "2. Biochar properties": You list a lot of data, but neither do you set them in context, nor do you explain a lot of the single results. Readers with less biochar background would feel lost. Exactly the readers who would read a review. Unfortunately, your oversimplified summary with the "general rule" and its "exceptions" does not reflect the (not so current) state of science in regard to biochar.
3. Biochar as a soil amendment
170: This is a very positive view on how biochar affects soil. However, this is only half-true, because these positive effects depend not only on (the myriad forms of) biochar, but also on the soil type, the investigated crop, how biochar was introduced into the soil, and many more factors. Based on this varying factors, biochar effects on soil can also be near indifferent or even negative. (which does not negate the high value of biochar for soil and humanity)
175: Please specify the non-supplemented soil. Poultry litter has very high nutrient contents, even after a lot of it is lost during pyrolysis (not the most effective way to utilize it). So what was compared, a non-fertilized with a fertilized soil? Which influence had the stable carbon (BC) in all this?
200: No, the "Increase in dry matter yield in this research shows that" only BC plus nutrients (and microorganisms I might add) improves the yield. By the way, this is the lesson from research of Terra Preta do Indio, from which the concept of biochar evolved.
213: Really, BC obtained from manure stimulated plant growth the most? Might that have something to do with the nutrients of the feedstock?
220: BC is not used as "nutrient addition", it is rather a mediator for nutrients (especially together with microorganisms), whether they were already in the feedstock or were (through various processes) added later.
229: Why did you review only five papers in this "very wide field of study"?
SUMMARY of "3. Biochar as a soil amendment": Again, a lot of data, though only five papers reviewed, without putting things in a wider context (it exists).
4. Biochar as a manure/sludge additive
249: Was the slurry spread on a field? The amount of slurry would be interesting.
255: The increase in CH4 would coincide with some studies that looked at improving biogas-production with BC additives (among others line 284).
273: Interesting author suggestion. What do you think? Is it more likely that BC was oxidized by, at most, 70°C, or could it be that the compost feedstock underwent a more intense composting (higher temperatures) an therefore more feedstock C was lost as CO2?
284: Amendment of BC to anaerobic digestion is an interesting field, but does it fit into this chapter? It might, but be aware of the irony! The increase of biogas production is kind of an increase of CH4 emissions, though here it is utilized and not an environmental problem (except for slight losses from the digestate and through gas leaks/incomplete combustion).
296: Reduction of odor emissions from composting is hardly mentioned in literature, because odor occurs only when the process fails. However, today composting is very well understood, at least how not to do it wrong. Non the less, BC can probably stabilize the process and make it even less likely that odor occurs. Anyway, it is not the best transition to BC's deodorizing potential in other areas.
SUMMARY of "4. Biochar as a manure/sludge additive": Quite interesting. Could there be any similarities to the function in soil. Putting things in context is still missing.
5. Biochar as a feed additive
406: Maybe the chain of primary use (feed additive), secondary use (fertilizer), and tertiary use (long-term C in soil) could be more emphasized?
SUMMARY of "5. Biochar as a feed additive": An area where I have the least knowledge. Therefore very informative. However, I wish that it would be tried to connect observed effects in animals with effects in other systems (soil, manure).
6. Conclusions
412: Since this paper is about alternative uses for BC, I would like to point to an old article from 2012 by Schmidt, "55 Uses of Biochar": http://www.ithaka-journal.net/55-anwendungen-von-pflanzenkohle?lang=en
SUMMARY of "6. Conclusions": The conclusions fit the rest of the paper, kind of ...there is this and that and one should try to find out more...
A lot of work went into this draft. However, right now it is rather a pure data collection, very interesting though. Some more work could turn it into a helpful and inspiring review.
Author Response
Please refer to the attachment: applsci-572576 reviewer 1.pdf

Reviewer 2 Report
Overview and general recommendation
Manuscript ‘A review of biochar properties and their utilization in crop agriculture and livestock production’ authored by Kalus, K., Koziel, J.A. and OpaliÅ„ski, S. provides an overview on biochar characteristics and agronomic applications, emphasizes the need for more research on biochar application as animal feed and provides a recommendation for future research.
Bearing in mind that the role of biochar as environmental management tool has been widely studied and that lately there is a growing interest in all the areas of biochar research and applications, the current manuscript represents an important review in the research field, focusing on the last 10 years. Besides, the manuscript is clearly written, well-structured and easy to read.
However, the main criticism is related to summary on the use of biochar as a soil amendment, as the authors considered quite a small portion of currently available literature. Despite a long and exhausting review is clearly not the purpose of this manuscript, there is a need to improve certain sections by providing more robust set of data and discussions while underlining the knowledge gaps. For these reasons the manuscript is not recommended for publishing in the current form, and major revisions are suggested.
My specific comments are listed below. I am kindly asking the authors to address these comments in their response.
Major comments
L55-160: Section ‘Biochar properties’ does not address other important properties such as: porosity, surface area, content of potentially toxic elements (PTEs) like metals, PAHs, etc. It is difficult to understand why the authors only decided to present certain biochar properties in this section. Moreover, there is no reference to two international biochar quality standards available - European Biochar Certificate and International Biochar Initiative Biochar Standards. This, however, is apparently important issue regarding biochar production and consequently, applications. Although the presence of contaminants in biochar may be overcome, to certain extent, with certification, these quality standards are voluntary. Perhaps this could be briefly mentioned in the current review. Additionally, not necessarily the total contaminant concentration in biochar is reflecting their bioavailability and/or fate in the environment. This is, of course, a rather complex issue, involving the factors such as biochar processing conditions, but also the context of biochar application – e.g. soil texture, etc. For instance, there is literature reporting this for bioavailability of biochar-bound PAHs to soil organisms. Therefore, in the context of biochar properties, PTEs should not be neglected, as there is a knowledge gap regarding their behaviour and bioavailability in biochar-amended soils.
L169 and onwards: Section 3 – ‘Biochar as a soil amendment’, as I mentioned previously, is represented with small number of studies, considering the currently available literature in this area of biochar research. Although I do agree that the authors point out the most relevant effects in this context, Table 2 provides a way too short summary and it does not seem to be representative of the literature published so far as focused on these effects. Therefore, the authors are advised to amend and revise this section accordingly. Also, please try to report the same unit for ‘biochar dose’ in Table 2. The units may be converted, e.g. % by mass into t ha-1.
L187-189: This idea is repeated more times, please try to avoid the repetition.
L194-196: The same as L187-189.
L304 and onwards: The section 5 might as well be updated with the latest published literature on the application of biochar in animal feeding.
L394-396: Could other parameters be e.g. PAHs, or some other organic contaminants? What about potential toxicity of biochar-bound contaminants and/or maybe particle size-related effects? These might be some limitations regarding the use of biochar as animal feed and the knowledge gaps that could be more elaborated, if possible.
Author Response
Please refer to the attachment: applsci-572576 reviewer 2.pdf

Round 2
Reviewer 1 Report
Dear authors,
Thank you for considering all previous comments. Compared to other related reviews, your manuscript is not very comprehensive, nor does it provide new findings based on the reviewed literature. However, it is a concise review that may serve as a good starting point for further study.
Reviewer 2 Report
Dear authors,
Thank you for addressing the comments and suggestions. I am satisfied with the revision performed.